# Structure of the mouse TRPC4 ion channel

Jingjing Duan[1,2], Jian Li[1,3], Bo Zeng [4], Gui-Lan Chen[4], Xiaogang Peng[5], Yixing Zhang[1], Jianbin Wang[1], David E. Clapham [2], Zongli Li[6] & Jin Zhang[1]

Members of the transient receptor potential (TRP) ion channels conduct cations into cells. They mediate functions ranging from neuronally mediated hot and cold sensation to intracellular organellar and primary ciliary signaling. Here we report a cryo-electron microscopy (cryo-EM) structure of TRPC4 in its unliganded (apo) state to an overall resolution of 3.3 Å. The structure reveals a unique architecture with a long pore loop stabilized by a disulfide bond. Beyond the shared tetrameric six-transmembrane fold, the TRPC4 structure deviates from other TRP channels with a unique cytosolic domain. This unique cytosolic N-terminal domain forms extensive aromatic contacts with the TRP and the C-terminal domains. The comparison of our structure with other known TRP structures provides molecular insights into TRPC4 ion selectivity and extends our knowledge of the diversity and evolution of the TRP channels.

[1] School of Basic Medical Sciences, Nanchang University, 330031 Nanchang, Jiangxi, China. [2] Howard Hughes Medical Institute, Janelia Research Campus, Ashburn, VA 20147, USA. [3] Department of Molecular and Cellular Biochemistry, University of Kentucky, Lexington, KY 40536, USA. [4] Key Laboratory of Medical Electrophysiology, Ministry of Education, and Institute of Cardiovascular Research, Southwest Medical University, 646000 Luzhou, Sichuan, China. [5] The Key Laboratory of Molecular Medicine, The Second Affiliated Hospital of Nanchang University, 330006 Nanchang, China. [6] Department of Biological Chemistry and Molecular Pharmacology, Howard Hughes Medical Institute, Harvard Medical School, Boston, MA 02115, USA. These authors contributed equally: Jingjing Duan, Jian Li, Bo Zeng. Correspondence and requests for materials should be addressed to Z.L. (email: zongli_li@hms.harvard.edu) or to J.Z. (email: zhangxiaokong@hotmail.com)

Mammalian transient receptor potential (TRP) channels are activated by a wide spectrum of ligands, temperature, lipids, pH, and as yet unknown stimuli. They are classified into six subfamilies based on sequence similarity: TRPC (canonical), TRPM (melastatin), TRPV (vanilloid), TRPA (ankyrin), TRPML (mucolipin), and TRPP (or PKD) (polycystin)[1]. The TRPC subfamily are non-selective cation channels ($Na^+$, $K^+$, $Ca^{2+}$) that alter proliferation, vascular tone, and synaptic plasticity[2,3]. This family can be further subdivided into two subgroups: TRPC2/3/6/7 and TRPC1/4/5. TRPC4 is broadly expressed in human tissues and can assemble as homomeric channels or form heteromeric channels with TRPC1 and TRPC5[4–7]. Studies of *Trpc4*-deficient mice have shown that TRPC4 affects endothelial-dependent regulation of vascular tone, endothelial permeability, and neurotransmitter release from thalamic interneurons[8]. Stimulation of $G_q$ and $G_{i/o}$ G-protein-coupled receptors (GPCRs) as well as tyrosine kinase receptors potentiate channel activity[9,10]. Activation of TRPC4 is regulated by intracellular $Ca^{2+}$, phospholipase C, and membrane lipids by unclear mechanisms. In addition, Storch et al.[11] have proposed a potential mechanism related to phosphatidylinositol 4,5-bisphosphate and $Na^+/H^+$ exchanger regulatory factor proteins[11].

Along with the revolution in cryo-electron microscopy (cryo-EM), improved sample preparation, data acquisition, and image processing strategies, the structures of TRPVs[12–17], TRPA1[18], TRPP1[19], TRPML1[20], TRPM4[21–24], and TRPM8[25] have been solved. Here we present the structure of mouse TRPC4 in its apo state at pH 7.5 at an overall resolution of 3.3 Å. The structure provides detailed information on the ion permeation, selectivity, and gating mechanism of TRPC subfamily.

## Results

**Overall structure of the mouse TRPC4 tetrameric ion channel.** The mouse TRPC4 (residues amino acids (a.a.) 1–758, excluding a.a. 759–974) was expressed using the BacMam expression system (Methods) and purified protein (pH 7.5) was used for single-particle cryo-EM analysis (Supplementary Fig. 1). The overall resolution of TRPC4 reconstruction was 3.3 Å (Supplementary Fig. 2 and Table 1), which enabled us to construct a near-atomic model (Supplementary Fig. 3). Disordered regions led to poor densities for 4 residues in the S1–S2 loop, 2 residues in the S3–S4 loop, 27 residues in the distal N terminus, and 28 residues in the truncated distal C terminus. In total, the TRPC4 structure is a four-fold symmetric homotetramer (Fig. 1a) with dimensions of 100 Å × 100 Å × 120 Å (Fig. 1b). Each monomer consists of a transmembrane domain (TMD) and a compact cytosolic domain. The cytosolic domain is composed of two subdomains: the N-terminal subdomain consisting of four ankyrin repeats (AR1–AR4) and seven α-helices (H1–H7), and the C-terminal subdomain containing a connecting helix and a coiled-coil domain (Fig. 1c, d).

$Ca^{2+}$ measurements and electrophysiological studies were performed to verify that the truncated construct used for structural investigation (residues a.a. 1–758, excluding a.a. 759–974) was permeable to cations and retained sensitivity to channel activator englerin A, blocker ML204, and blocker 2-APB (Fig. 2). Englerin A induced a robust rise of intracellular $Ca^{2+}$ in both full-length and truncated TRPC4-transfected 293T cells, but not in the empty vector-transfected control cells (Fig. 2a). In patch clamp experiments, englerin A potentiated whole-cell currents in a dose-dependent manner for 293T cells expressing full-length and truncated TRPC4. In addition, currents were inhibited by the TRPC4 blocker ML204 and the non-selective blocker 2-APB (Fig. 2b). The current–voltage (I–V) relationships from both constructs were typical of TRPC4/5, with flattening of

the curve between 10 and 40 mV due to outward $Mg^{2+}$ block[26]. These results suggest that, in the context of chemical modulation, the biophysical properties of the truncated construct are similar to that of the full-length TRPC4.

To futher test the functional properties of the truncated construct, we examined receptor-operated activation of TRPC4 by coexpression with GPCRs. In cells transfected with TRPC4 constructs and P2Y1 and P2Y2 receptors (both coupled to $G_q$ proteins), extracellular application of ADP (P2Y1 agonist) or ATP (P2Y2 agonist) induced large currents for full-length TRPC4. In contrast, the currents for truncated TRPC4 were much smaller but still exhibited the characteristic I–V relationship of the TRPC4 channel (Supplementary Fig. 4a). The truncated TRPC4 did not respond to $G_{i/o}$ receptor agonists DAMGO (μOR) or carbachol (M2R) (Supplementary Fig. 4b).

**Major structural differences with other TRP subfamilies.** In Fig. 3, we compare the TRPC4 structure with previously reported TRP structures. Not surprisingly, the organization of six helices in each TMD is similar to that of other TRP channels, while the intracellular architecture is distinct. By superimposing a TRPC4 monomer with representative TRP monomers from each subfamily, we found that the overall fold of TRPC4 is closest to that of TRPM4 (Fig. 3a). TRPC4 has marked similarities to TRPM4 in the TMDs despite their different tissue functions and lack of sequence conservation (<20% identical residues) (Supplementary Fig. 5a, b). Distinctive features of TRPC4 include: (1) the

### Table 1 Data collection and refinement statistics

| Cryo-EM of TRPC4 | [EMD:6901] [PDB:5Z96] |
|---|---|
| **Data collection and processing** | |
| Microscope | FEI Tecnai Polara |
| Detector | Gatan K2 |
| Calibrated magnification | 40,607 |
| Voltage (kV) | 300 |
| Electron exposure (e⁻/Å²) | 52.8 |
| Defocus range (μm) | 1.5–3.0 |
| Pixel size (Å) | 1.23 |
| Symmetry imposed | C4 |
| Initial particle images (no.) | 381,165 |
| Final particle images (no.) | 232,858 |
| FSC threshold | 0.143 |
| Map resolution (Å) | 3.3 |
| **Refinement** | |
| Refinement software | phenix.real_space_refine |
| Initial model used (PDB code) | De novo |
| Model composition | |
| Non-hydrogen atoms | 21,506 |
| Protein residues | 2608 |
| Ligands | 12 |
| *B* factors (Å²) | |
| Average | 65.46 |
| r.m.s. deviations | |
| Bond lengths (Å) | 0.01 |
| Bond angles (°) | 1.11 |
| Validation | |
| MolProbity score | 1.83 |
| Clashscore | 7.25 |
| Ramachandran plot | |
| Favored (%) | 97.39 |
| Allowed (%) | 2.08 |
| Disallowed (%) | 0.53 |

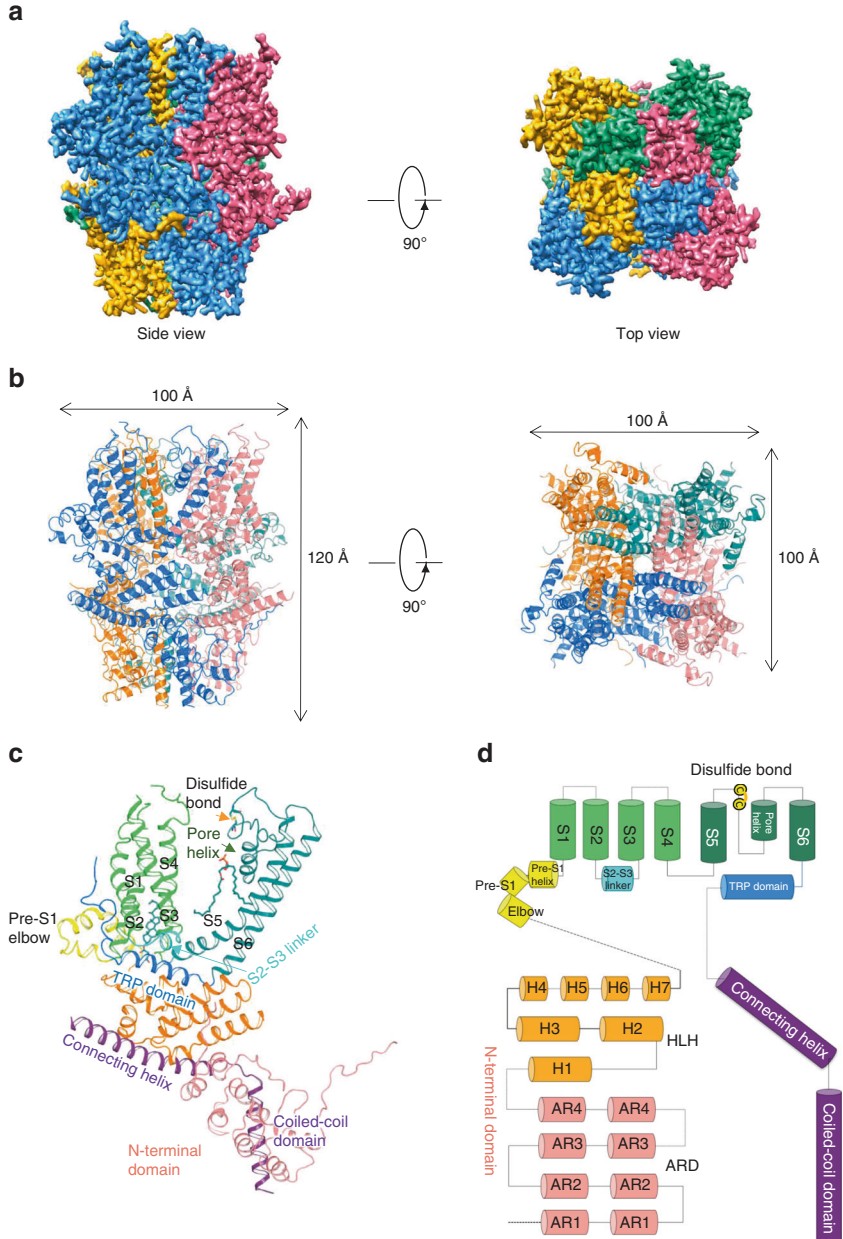

**Fig. 1** Overall structure of mTRPC4. **a** Side and top views of the cryo-EM density map of mouse TRPC4 at 3.3 Å overall resolution. Each monomer is represented in different colors. **b** Ribbon diagrams of the mouse TRPC4 model with channel dimensions indicated. **c** Ribbon diagrams depicting structural details of a single subunit. **d** Linear diagram depicting the major structural domains of the TRPC4 monomer, color-coded to match the ribbon diagram in (**c**)

arrangement of S2–S3 linker, S5, S6, and the pore loop. In TRPC4, the S2–S3 linker has two-helical turns, shorter than that of TRPM4 (Supplementary Fig. 5a), which limits the interactions of S2 and S3 with their cytoplasmic regions; (2) the disulfide bond between TRPC4's Cys549 and Cys554 lies in the loop linking S5 and the pore helix (Fig. 3b, c), while TRPM family's disulfide bond is located in the loop between the pore helix and S6[24] (Supplementary Fig. 5c). Note that these two cysteines are conserved in TPRC1/4/5, but not in other TRPC members (Supplementary Fig. 5b); (3) a pre-S1 elbow helix connects the N terminus and TMD in TRPC4 (Fig. 3d), as in TRPM4 and NOMPC (no mechanoreceptor potential C)[21,27]; however, TRPC4 and TRPM4's pre-S1 helix is not found in NOMPC[27]. In TRPC4 the pre-S1 elbow helix is longer, bending and connecting to the pre-S1 helix directly, while in TRPM4 a

characteristic bridge loop (approximately 60 residues) connects the pre-S1 helix with the pre-S1 elbow (Fig. 3d and Supplementary Fig. 5c).

To understand the role of the disulfide bond in the pore loop of TRPC4, we performed patch clamp experiments on wild-type (wt) and cysteine mutants of TRPC4. Reducing agents dithiothreitol (DTT; membrane permeable) and tris(2-carboxyethyl) phosphine hydrochloride (TCEP; membrane impermeable) potentiated wt TRPC4 whole-cell currents (Fig. 3e). The $I–V$ relationship of the DTT-induced current was characteristic of TRPC4, while that induced by TCEP was similar to the $I–V$ commonly seen for other TRP channels such as TRPV1/M7/M8. This may be due to its stronger reducing effect and exclusive extracellular action. The double cysteine mutant (C549A+C554A) could be activated by englerin A but not DTT

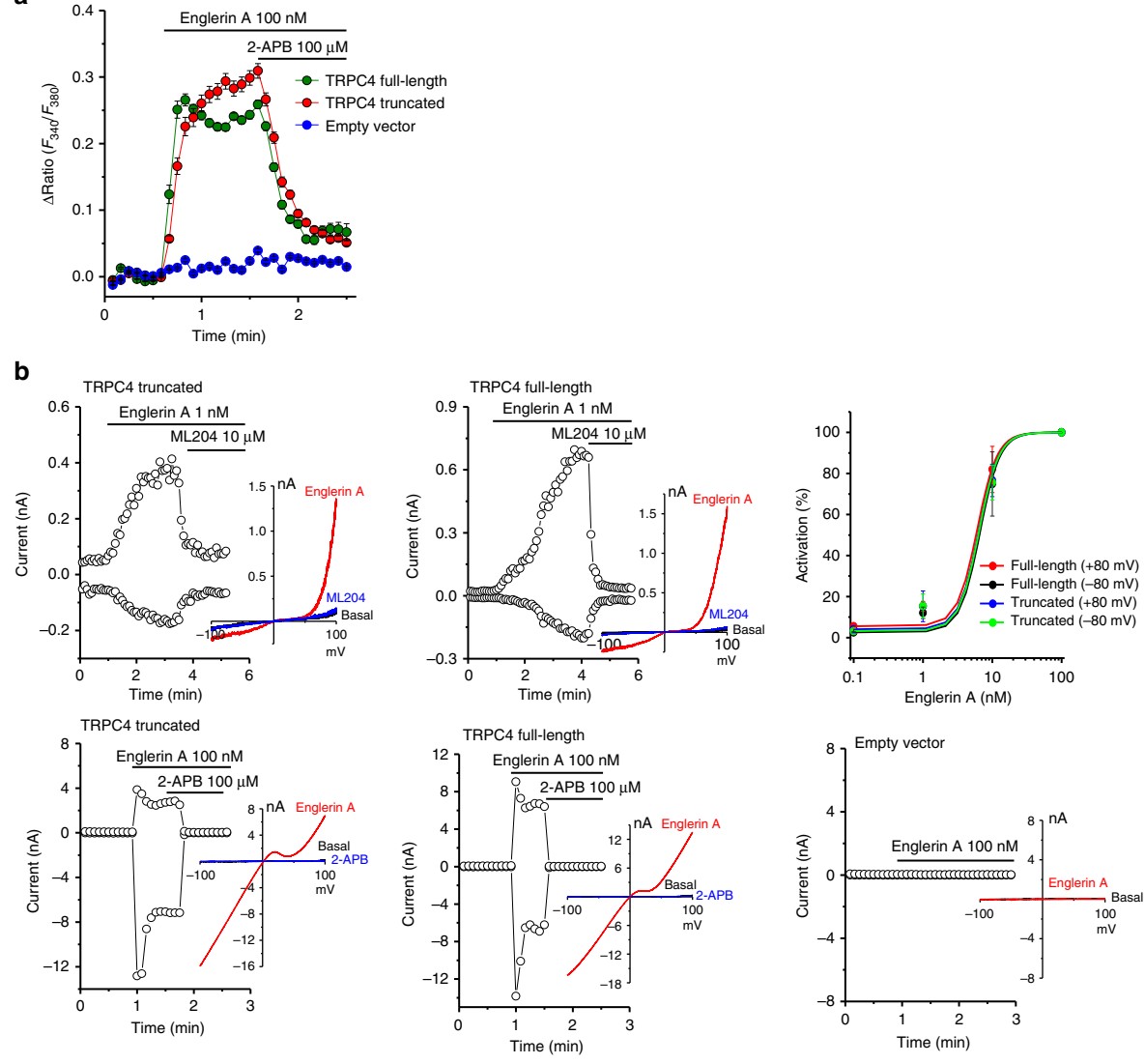

**Fig. 2** The truncated TRPC4 construct used in cryo-EM encodes a functional channel. **a** Intracellular $Ca^{2+}$ measurements of truncated mTRPC4, full-length mTRPC4, and empty vector-transfected 293T cells loaded with Fura-2 AM. Data are expressed as mean ± s.e.m.; $n = 10$ cells in each group. **b** Representative whole-cell patch clamp recordings and $I-V$ relationships of truncated mTRPC4, full-length mTRPC4, and empty vector expressed in 293T cells. The time course of currents measured at +80 and −80 mV and $I-V$ relationships of the peak currents from different conditions are shown. The upper right panel shows dose–response curves of englerin A in activating TRPC4 currents at +80 and −80 mV (values with error bars are expressed as mean ± s.e.m.; $n = 8$ for each concentration). Current induced by 100 nM englerin A in each cell was considered as maximum current (100%); currents induced by other concentrations of englerin A in the same cell were normalized. Englerin A is a TRPC4/5 activator and ML204 is an inhibitor for TRPC4/5 channels. 2-APB is a non-selective blocker of TRP channels

(Fig. 3f), suggesting that the TRPC4 channel without the pore loop disulfide bond is still functional but lacks redox sensitivity. Surprisingly, mutation of a single cysteine (C549A or C554A) resulted in insensitivity to englerin A and DTT (Fig. 3g). If this mutated TRPC4 trafficked to the plasma membrane, the loss of englerin A and DTT sensitivities suggests that the channel's pore loop architecture has been severely disrupted, resulting in a channel that cannot be exogenously activated.

**Cytosolic domain features and interactions**. The cytosolic domains of TRP channels include regulatory components and domain interactions that may tune channel gating. The cytosolic domain of TRPC4 adopts a pedestal-like architecture (Fig. 4a and Supplementary Fig. 6a). The large and unique N-terminal domain

of TRPC4 contains a long loop followed by an ankyrin repeat domain and helix–loop–helix (HLH) motifs. These HLH motifs consist of seven helices and several connecting loops (Fig. 1c, d and Supplementary Fig. 6a). Similar to TRPM structures, the C-terminal domain of TRPC4 is composed of two helices, a connecting helix and a coiled-coil domain helix (Fig. 1c, d). The connecting and coiled-coil domain helices bend ~120° to form an inverted "L" (Supplementary Fig. 6b). The coiled-coil domain contains three heptad repeats that exhibit the characteristic periodicity $(a-b-c-d-e-f-g)_n$ (Fig. 4b, c and Supplementary Fig. 7), with hydrophobic residues at positions "a" and "d". The presence of Val and Ile at the "a" position, and Leu and Phe at the "d" position in the core of the coiled-coil domain supports the formation of a tetramer (Fig. 4c).

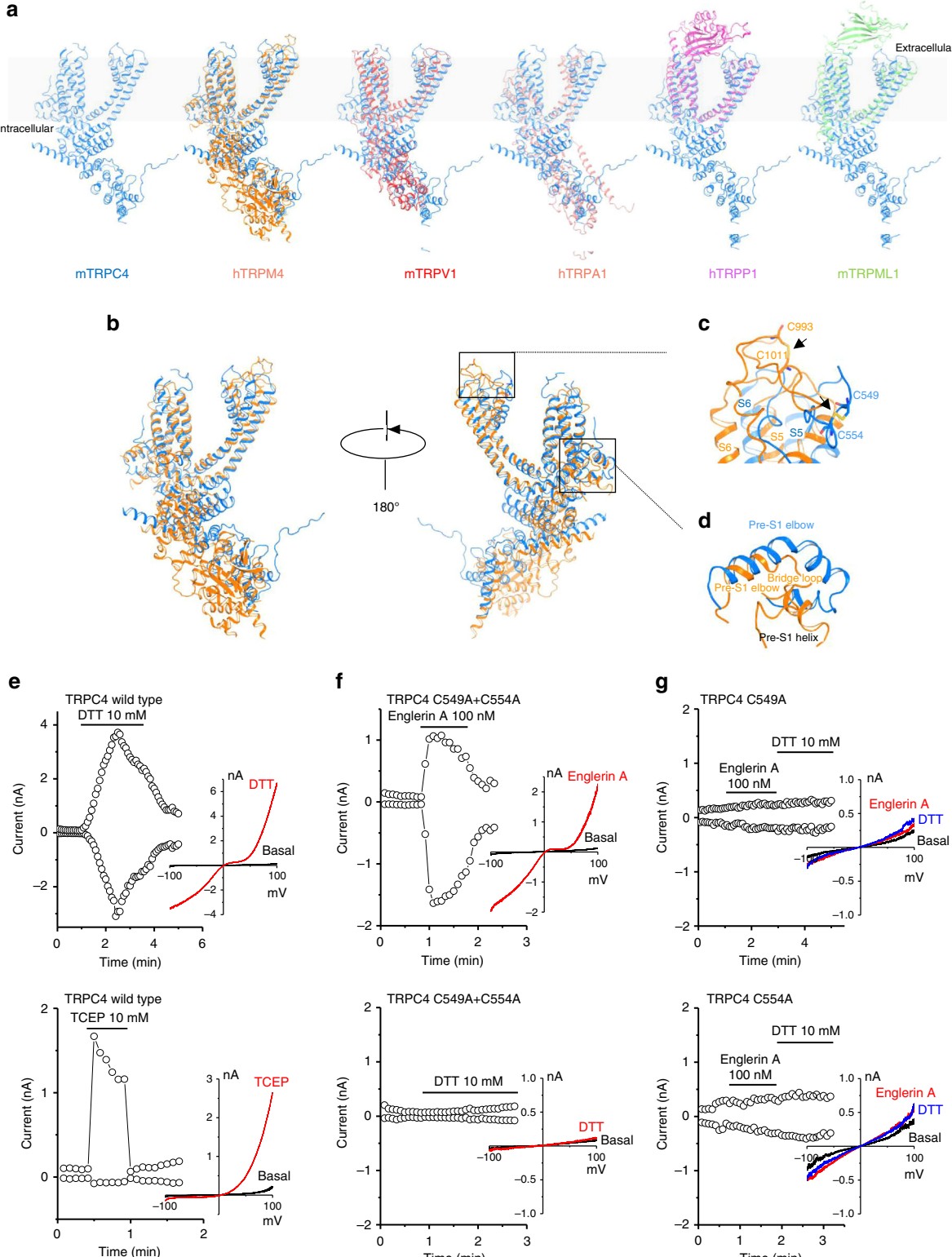

**Fig. 3** Comparison of the TRPC4 structure with previously solved TRP channel structures (apo state). **a** Side views of an mTRPC4 subunit compared with other TRP family members including human TRPM4 [PDB:6BWI][24], mouse TRPV1 [PDB:3J5P][14], human TRPA1 [PDB:3J9P][18], human PKD2/TRPP1 [PDB:5T4D][19], and mouse TRPML1 [PDB:5WPV][20]. **b** Superimposition of TRPC4 and TRPM4. **c** Key pore loop disulfide bond between Cys549 and Cys554 in TRPC4 and the corresponding pore loop disulfide bond between Cys993 and Cys1011 in TRPM4 (black arrows). **d** Differences in the organizations of the linker (pre-S1 elbow and pre-S1 helix) between the N terminus and transmembrane domains in TRPC4 and TRPM4. **e** Activation of whole-cell currents by reducing agents DTT and TCEP in 293T cells overexpressing wild-type TRPC4. **f** The double cysteine TRPC4 mutant (C549A+C554A) is activated by englerin A but not DTT. **g** Single cysteine mutants, C549A or C554A, are insensitive to 100 nM englerin A and 10 mM DTT. The time course of currents measured at +80 and −80 mV and I–V relationships of the peak currents in different conditions are shown

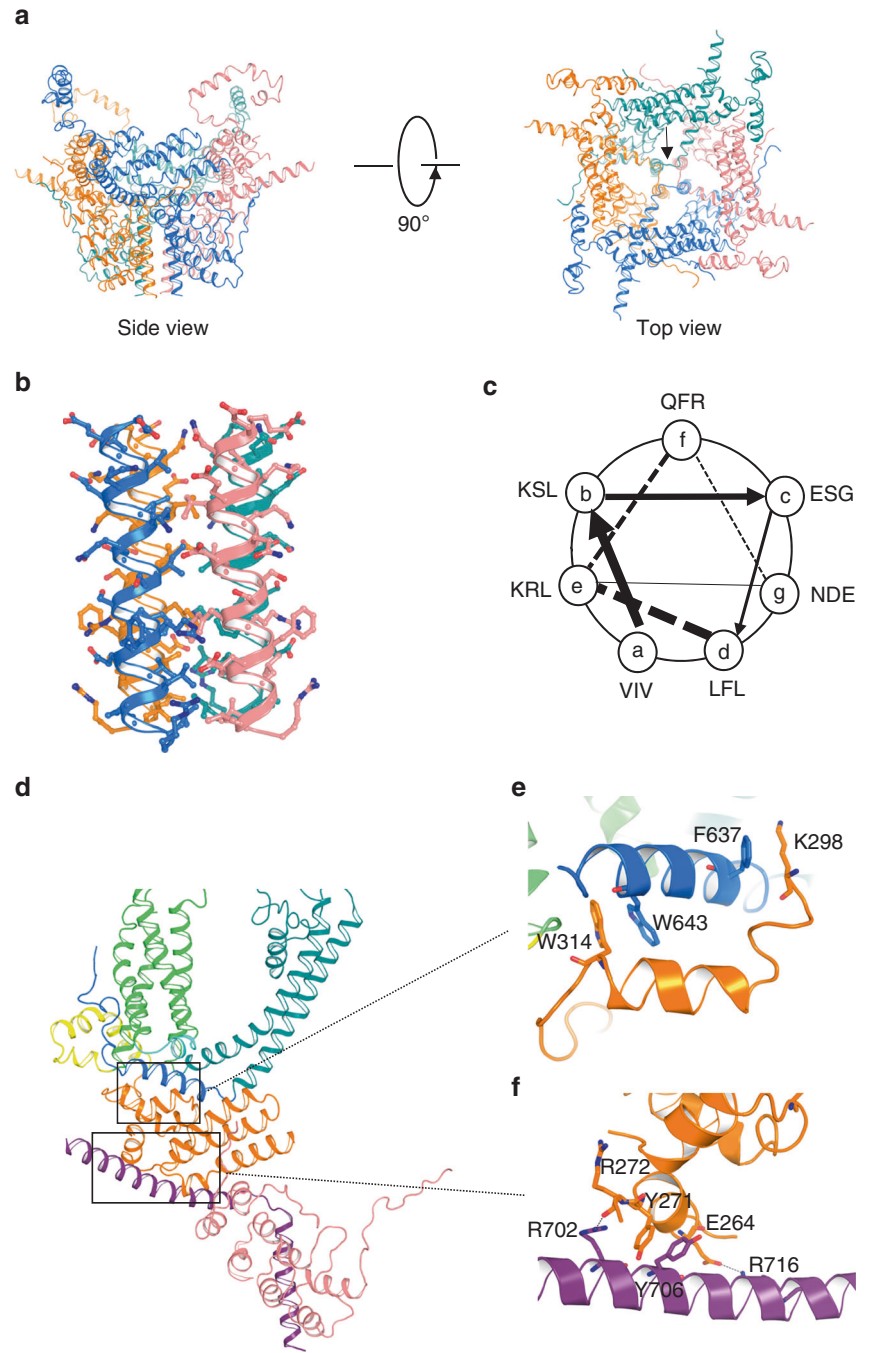

**Fig. 4** Unique cytosolic domains and interactions. **a** Side and top views of the cytosolic domains including full-length N- and truncated C-terminal domains. **b** Ribbon diagram of the tetrameric coiled-coil domain structure. Side chains are represented by ball and stick models. **c** Helical wheel projection of the residues in the coiled-coil domain of TRPC4. **d**–**f** Side views of a single subunit of the N-terminal domain to illustrate the locations of the interactions between the (**e**) TRP domain (blue) and N-terminal domain (orange) and (**f**) N-terminal domain (orange) and truncated C-terminal domain (purple)

Aromatic interactions are important in cytosolic domain arrangements and protein folding[28]. The TRP domain and N-terminal domain interactions are stabilized by π–π interactions (formed by Trp643 with Trp314) and cation–π interactions (formed by Phe637 with Lys298; Fig. 4d, e). The N-terminal and C-terminal domains interface is also strengthened by a π–π interaction (Tyr271 with Tyr706) and two hydrogen bonds (Glu264 with Arg716, Arg272 with Arg702) (Fig. 4d, f).

**Ion conduction pore and binding sites for cation and lipids.** Positioned C terminal to the pore helix, Gly577 marks a restriction point of 6.7 Å between diagonally opposed residues (Fig. 5a, b). The corresponding filter-forming residue in TRPM4 is Gly976 at a 6.0 Å constriction. Compared to TRPM4, TRPC4's selectivity filter is slightly more open, but the ion conduction pathway is restricted at its cytoplasmic interface, with Ile617, Asn621, and Gln625 at the bottom of S6 defining a lower gate. The narrowest constriction of the ion conduction pathway (3.6 Å) is formed by the S6 side chains of Asn621 (Fig. 5c and Supplementary Fig. 8a), while in TRPM4, the 5.1 Å wide lower gate is positioned at Ile1040 (Supplementary Fig. 8b). In contrast, the most restricted point in TRPV1 is in the selectivity filter (4.8 Å)

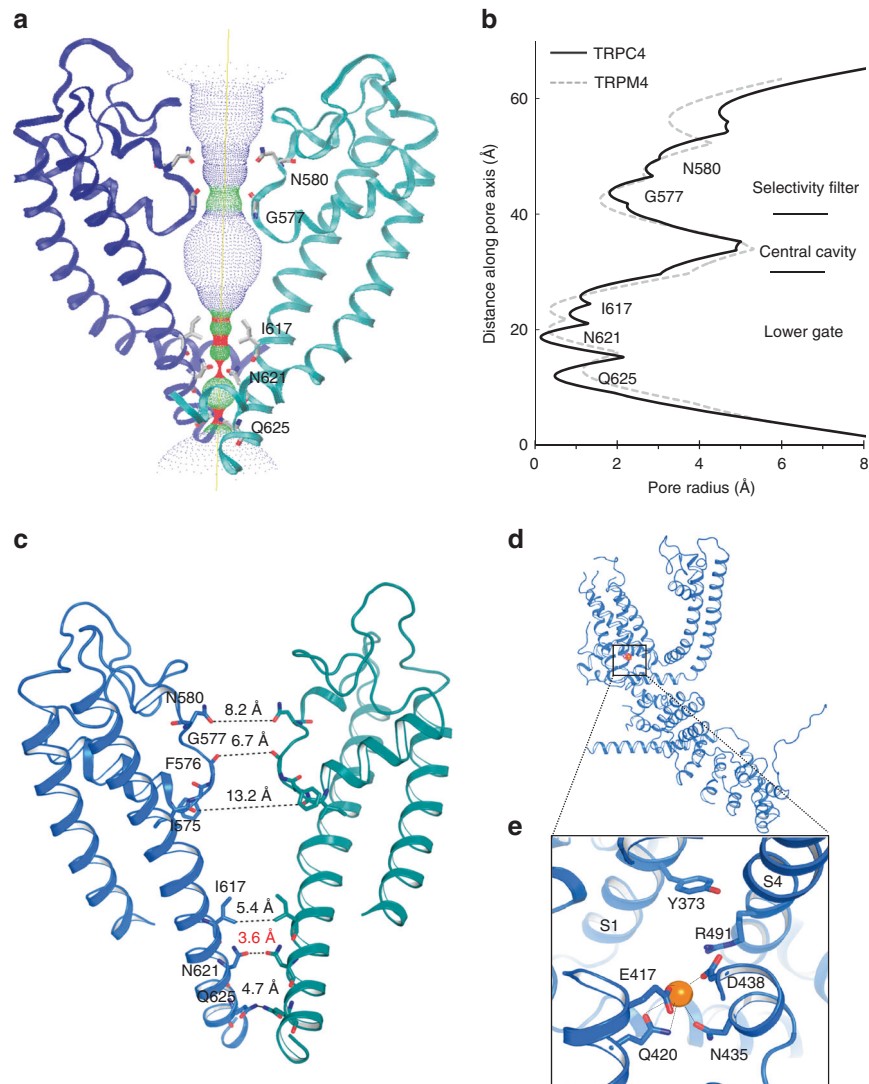

**Fig. 5** TRPC4 ion conduction pathway. **a** Ion conduction pathway shown as dots and mapped using HOLE. **b** Pore radius along the central axis. The side chains of G577 form a narrow constriction at the selectivity filter. N621 is the most restricted site of the lower gate. The dashed line represents TRPM4 for comparison. **c** Side view of TRPC4's pore region with chains A and C. The distances between diagonal residues in the selectivity filter and lower gate are labeled. **d** A putative Na+ binding site is found on the cytosolic side in the hydrophilic pocket of the S1–S4 domain. **e** Enlarged view of putative Na+ (orange sphere) ion interacting with E417, Q420, N435, and D438

between opposing Gly643 residues (Supplementary Fig. 8c)[29]. In TRPA1, the narrowest point (6.1 Å) is Val961 at its lower gate[18] (Supplementary Fig. 8d). These ~0.5–2.5 Å differences in the narrowest point of TRPs structures may give some clue as to ion selectivity, but we also are aware that current resolution optimization in cryo-EM is still being improved by methods such as model-based local density sharpening[30], and resolution varies with location within the particle, conditions such as vitrification, and electron density map fitting.

The simplest hypothesis, with these caveats in mind, is that TRPC4 is in a closed or inactivated state since the lower gate is too narrow to allow the passage of a fully or partially hydrated ion. In support of this idea is the fact that Gln625 (located in the ion conduction exit pathway) is conserved in all the TRPC channels (Supplementary Fig. 9), suggesting it plays an important role in ion permeation.

TRPC4 is non-selective and thus permeable to monovalents (Na+, K+) and some divalents, such as Ca2+. A strong non-protein density peak in our TRPC4 structure is present in a hydrophilic pocket on the cytoplasmic side of the S1–S4 fold,

consistent with the corresponding location of a presumed Ca2+ in TRPM4[21] (Fig. 5d, e and Supplementary Fig. 10). We tentatively modeled this non-protein density as Na+ since sodium was the most abundant cation in our purification buffer. The assumed Na+ located at the cytoplasmic face is apparently coordinated by side chains of Glu417 and Gln420 from S2 and the Asp438 and Asn435 from S3 (Fig. 5e). The negatively charged Glu417 and Asp438 are conserved within the TRPC subfamily (except TRPC1) (Supplementary Fig. 9). S1's Tyr373 and the positively charged S4 Arg491 are located above the cation binding site, forming a lid that may prevent the outward movement of cations (Fig. 5e).

Eight densities corresponding to lipid molecules were clearly resolved and identified as cholesteryl hemisuccinates (CHS) and phospholipids (the density fitting ceramide-1-phosphate or phosphatidic acid) (Supplementary Figs. 3 and 11). Four CHS located at the interface of the N-terminal domain and the S4/S5 linker are bound to each protomer, stabilizing the domain interaction (Supplementary Fig. 11). The phospholipid is embedded in the gap between the four monomeric subunits with

its polar head interacting with the pore helix and neighboring S6 helix (Supplementary Fig. 11). In vivo phosphorylation or dephosphorylation of membrane lipids could thus alter the topology of the ion conduction pathway.

## Discussion

TRPC4 has been characterized as a non-selective cation channel with moderate selectivity for $Ca^{2+}$ over monovalent cations. Comparison of this TRPC structure with other TRP channel structures highlights some commonalities and differences. All TRP channels are tetramers with domain swapping interactions, pore loops, selectivity filters, and extracellular and intracellular-facing constriction sites, as first shown for six TM $K^+$ channels[31]. However, the lower gate in the TRPC4 appears to have an unusual set of three constriction sites not found in other TRP channel structures. Another interesting feature that bears functional investigation is the extracellular pore loop disulfide bond (e.g., TRPC4 and TRPM4).

Our structure of TRPC4 provides insights into the architecture of the selectivity filter and lower gate. Mutations within this pore region leads to changes in ion permeability; however, mutagenesis of the pore-localized LFW motif (a.a. 571–573 in TRPC4), which is conserved in all TRPC members, results in a dominant-negative channel[4]. This phenomenon can be explained in our structure, by the pore helix located at the LFW motif (which has a π–π interaction; Supplementary Fig. 12a, b). This mutation likely perturbs the stability of the key pore region, leading to protein misfolding and subsequent degradation. Notably, mutation of Gly503 to serine resulted in spontaneously active channels which could not be further activated by receptor and intracellular $Ca^{2+}$[32]. In our TRPC4 structure, Gly503, which is located on the S4/S5 linker, forms a hydrogen bond with Trp635 on the TRP domain, suggesting that the S4/S5 linker and TRP domain interact to regulate channel activity[32] (Supplementary Fig. 12c).

The role of the disulfide bond in TRPC4's pore loop has not been previously investigated, but electrophysiological studies on the closely related TRPC5 have shown that reduction of the disulfide bond with DTT potentiated TRPC5 currents[33,34]. However, the results of TRPC5 cysteine mutations are disparate. Xu et al.[33] reported that mutation of either a single cysteine (C553A, C558A, or C553S) or both cysteines (C553A+C558A) resulted in constitutively active TRPC5 channels. Moreover, the disulfide bond was formed between C553 and C558 in the same monomer and not crosslinked between different monomers. These observations were challenged by a later study stating that both C553S and C558S mutants completely lost channel activity, and dimerization of channel proteins was significantly impaired for cysteine mutants of both TRPC4 and TRPC5[34].

Our structural results demonstrate the existence of a disulfide bond in the pore region. If one of the two cysteines is mutated, an inter-monomer disulfide bond may be formed which would substantially change the architecture of the pore loop and affect channel activity. This is supported by our electrophysiological observations from the C549A and C554A TRPC4 mutants. Specifically, results of the double cysteine mutant (C549A+C554A) suggest that the pore loop disulfide bond is dispensable for channel activation by the activator, but is essential for redox regulation of channel activity.

TRPCs have multiple activation mechanisms. TRPC1/4, 1/5 heterotetramers and TRPC4 or 5 homotetramers can be activated by $G_q$ protein-coupled receptors and increasing intracellular $Ca^{2+}$ as well as by $PIP_2$ hydrolysis, while TRPC3, 6, and 7 are additionally sensitive to diacylglycerol[35]. From our observations, we propose that our TRPC4 structure represents a closed state because of the narrow distance between the residues in the lower gate. In the present study, we truncated a.a. 759–974 from TRPC4, resulting in the absence of the PDZ-binding motif, a $G_\alpha$-binding sites, the second camodulin-binding domain and a potential $PIP_2$-binding region. We found that this truncation greatly impaired or abolished channel activity in response to $G_q$-mediated and $G_{i/o}$-mediated signaling, which confirmed the importance of the C terminus in receptor-operated activation of TRPC4. However, the truncated TRPC4 currents stimulated by P2Y1 and P2Y2 were small and thus there are other structural constraints related to $G_q$-mediated TRPC4 activation. The truncated TRPC4 retained the first camodulin-binding domain and thus could still be sensitive to intracellular $Ca^{2+}$ changes, such $Ca^{2+}$ release from the ER stores induced by $IP_3$. Our current study should help guide future mutagenesis, functional, and structural studies.

## Methods

**Protein expression and purification.** The mouse TRPC4 construct (a.a. 1–758 of 974) was cloned into the pEG BacMam vector[36] and a maltose-binding protein (MBP) tag was added to its N terminus (all primer sequences used in this study are in a Supplementary Table 1). P3 baculovirus were produced in the Bac-to-Bac Baculovirus Expression System (Invitrogen). HEK293S GnTI⁻ cells (from ATCC) were infected with 10% (v/v) P3 baculovirus at a density of $2.0–3.0 \times 10^6$ cells/ml for protein expression at 37 °C. After 12–24 h, 10 mM sodium butyrate was added and the temperature reduced to 30 °C. Cells were harvested at 72 h after transduction, and resuspended in a buffer containing 30 mM HEPES, 150 mM NaCl, 1 mM DTT, pH 7.5, with EDTA-free protease inhibitor cocktail (Roche). After 30 min, cells were solubilized for 2–3 h in a buffer containing 1.0% (w/v) N-dodecyl-β-D-maltopyranoside (Affymetrix), 0.1% (w/v) CHS (Sigma), 30 mM HEPES, 150 mM NaCl, 1 mM DTT, pH 7.5, with EDTA-free protease inhibitor cocktail (Roche). The supernatant was isolated by $100,000 \times g$ centrifugation for 60 min, followed by incubation in amylose resin (New England BioLabs) at 4 °C overnight. The resin was washed with 20 column volumes of "wash buffer" containing 30 mM HEPES, 150 mM NaCl, 0.1% (w/v) digitonin, 0.01% (w/v) CHS, 1 mM DTT, pH 7.5, with EDTA-free protease inhibitor cocktail (Roche). The protein was eluted with four column volumes of wash buffer with 40 mM maltose. The protein was then concentrated to 0.5 ml with a 100 kDa molecular weight cut-off concentrator (Millipore). PreScission protease was added to the samples and incubated overnight at 4 °C to remove the MBP tag. After incubation at 4 °C overnight, the protein was then purified on a Superose 6 column in a buffer composed of 30 mM HEPES, 150 mM NaCl, 0.1% (w/v) digitonin, 1 mM DTT, pH 7.5. The peak corresponding to tetrameric TRPC4 was collected and concentrated to 4.5 mg/ml for cryo-EM studies.

**EM data collection.** Purified TRPC4 protein (3.5 μl) in digitonin at 4.5 mg/ml was applied onto a glow-discharged, 400 mesh copper Quantifoil R1.2/1.3 holey carbon grid (Quantifoil). Grids were blotted for 7 s at 100% humidity and flash frozen by liquid nitrogen-cooled liquid ethane using a FEI Vitrobot Mark I (FEI). The grid was then loaded onto an FEI TF30 Polara electron microscope operated at 300 kV accelerating voltage. Image stacks were recorded on a Gatan K2 Summit (Gatan) direct detector set in super-resolution counting mode using SerialEM[37], with a defocus range between 1.5 and 3.0 μm. The electron dose was set to 8 e⁻/physical pixel/s and the sub-frame time to 200 ms. A total exposure time of 10 s resulted in 50 sub-frames per image stack. The total electron dose was 52.8 e⁻/Å² (~1.1 e⁻/Å² per sub-frame).

**Image processing and 3D reconstruction.** Image stacks were gain normalized and binned by 2× to a pixel size of 1.23 Å prior to drift and local movement correction using motionCor2[38]. The images from the sum of all frames with dose weighting were subjected to visual inspection and poor images were removed before particle picking. Particle picking and subsequent bad particle elimination through 2D classification was performed using Python scripts/programs[39] with minor modifications in the 8× binned images. The selected 2D class averages were used to build an initial model using the common lines approach implemented in SPIDER[40] through Maofu Liao's Python scripts[39], which was applied to later 3D classification using RELION[41]. Contrast transfer function parameters were estimated using CTFFIND4[42] using the sum of all frames without dose weighting. Quality particle images were then boxed out from the dose-weighted sum of all 50 frames and subjected to RELION 3D classification. RELION 3D refinements were then performed on selected classes for the final map[43]. The resolution of this map was further improved by using the sum of sub-frames 1–14.

**Model building, refinement, and validation.** For the TRPC4 structure, a poly-alanine model was first built in COOT[44]. Taking advantage of the defined geometry of helices and clear bumps for Cα atoms in the TMD, amino acid assignment was subsequently achieved based primarily on the clearly defined side chain densities of

bulky residues. The refined atomic model was further visualized in COOT. A few residues with side chains moving out of the density during the refinement were fixed manually, followed by further refinement. The TRPC4 model was then subjected to global refinement and minimization in real space using the PHENIX[45] module "phenix.real_space_refine"[46] and geometry of the model was assessed using MolProbity[47] in the comprehensive model validation section of PHENIX. The final model exhibited good geometry as indicated by the Ramachandran plot (preferred region, 97.39%; allowed region, 2.08%; outliers, 0.53%). The pore radius was calculated using HOLE[48].

**Electrophysiology and Ca$^{2+}$ measurements**. Mouse TRPC4α, TRPC4β, and μOR were cloned from mouse brain complementary DNA (cDNA) (primer sequences used in this study are in Supplementary Table 1). Human P2Y1 and P2Y2 were cloned from cDNA of HEK293T/17 cells (from ATCC). The human M2R construct was purchased from Cyagen (Guangzhou, China). C549A, C554A, and C549A+C554A mutants were generated by mutagenesis on mTRPC4α. mTRPC4β was used as the full-length control to assess the functionality of the truncated construct used for cryo-EM analysis. The TRPC4/GPCR constructs or empty vector were transfected into 293T cells together with an mCherry plasmid. Cells with red fluorescence were selected for whole-cell patch recordings (HEKA EPC10 USB amplifier, Patchmaster 2.90 software). A 1-s ramp protocol from –100 to +100 mV was applied at a frequency of 0.2 Hz. Signals were sampled at 10 kHz and filtered at 3 kHz. The pipette solution contained (mM): 140 CsCl, 1 MgCl$_2$, 0.03 CaCl$_2$, 0.05 EGTA, 10 HEPES, and the pH was titrated to 7.2 using CsOH. The standard bath solution contained (mM): 140 NaCl, 5 KCl, 1 MgCl$_2$, 2 CaCl$_2$, 10 HEPES, 10 D-glucose, and the pH was adjusted to 7.4 with NaOH. The recording chamber had a volume of 150 μl and was perfused at a rate of ~2 ml/min. For Ca$^{2+}$ imaging experiments, transfected 293T cells were seeded on coverslips and incubated with Fura-2 AM (2 μM) for 30 min at 37 °C in standard bath solution. The ratio ($F_{340}/F_{380}$) of Ca$^{2+}$ dye fluorescence was measured by a Nikon Ti-E system with the NIS-Elements software. All the experiments were performed at room temperature.

**Data availability**. Data supporting the findings of this manuscript are available from the corresponding authors upon reasonable request. Data deposition: Cryo-EM electron density map of the mouse TRPC4 has been deposited in the Electron Microscopy Data Bank, [EMD:6901] https://www.ebi.ac.uk/pdbe/emdb/, and the fitted coordinate has been deposited in the Protein Data Bank, [PDB:5Z96] www.pdb.org.

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

## Acknowledgements

We thank Steve Harrison and the Cryo-EM Facility (Harvard Medical School) for use of their microscopes. We also thank Maofu Liao for providing the Python scripts and help in image processing, and Corey Valinsky's for help on manuscript revision. J.Z. was supported by the Thousand Young Talents Program of China and National Natural Science Foundation of China (Grant No. 31770795). J.L. was supported by the National Natural Science Foundation of China (Grant No. 81402850). Functional studies in this project were supported by the National Natural Science Foundation of China (31300949 to B.Z. and 31300965 to G.-L.C.)

## Author contributions

J.Z. and J.D. designed and made constructs for BacMam expression and determined the condition to enhance protein stability. J.Z. purified the protein. Z.L. carried out detailed cryo-EM experiments, including data acquisition and processing. J.L. and J.Z. built the atomic model on the basis of cryo-EM maps. B.Z. and G.-L.C. performed functional studies. X.P., Y.Z., and J.W. assisted with protein purification and the mutation of TRPC4 constructs for functional studies. J.D., D.E.C. and J.Z. drafted the initial manuscript. All authors contributed to structure analysis/interpretation and manuscript revision. J.Z. and Z.L. initiated the project, planned and analyzed experiments, and supervised the research.

## Additional information

**Competing interests:** The authors declare no competing interests.

