## [Peer Review File · Nature Communications]

Reviewers' comments:

Reviewer #1 (Remarks to the Author):

In the present manuscript, J. Duan and colleagues report the first structure of the murine TRPC4 channel at a quite high resolution of 3.3 Å. While structures of members of all other TRP channel subfamilies have already been published, a TRPC structure is still lacking. In principle, this manuscript is of outstanding interest for the scientific community.

In the meantime, the submitted manuscript and one additional TRPC4 structure and structures of TRPC3 and TRPC6 are deposited on the bioRxiv server and publicly accessible, reflecting the astounding pace with which TRP channel cryo-EM structures are produced at present.

I have only a few minor comments which may further benefit the readability of the present paper such that a broad scientific audience can profit from it.

Specific points:

(1) Abstract: Please explain what is meant by "apo" state. Maybe the authors could write: "... in its unliganded (apo) state ...".

(2) Abstract, line 29: Please start new sentence: "The unique ...".

(3) Introduction, line 47: Please also cite Storch et al., Proc Natl Acad Sci U S A. 2017 Jan 3;114(1):E37-E46. doi: 10.1073/pnas.1612263114. A potential mechanism underlying lipid regulation is proposed in here.

(4) Supplementary Fig. 1b: Why were whole-cell currents in Suppl. Fig. 1b induced with trypsin? The truncated receptor inactivates within a few seconds, while the full-length channel stays activated for more than 2 min. Any explanation for this phenomenon? Can one really rule out different biophysical properties caused by the truncations?

(5) Figure 2: Please describe the experiments shown in Fig. 2 in the text in more detail. The reader is completely left alone to interpret the experiments shown.

My recommendation would be: Minor revision

Reviewer #2 (Remarks to the Author):

TRP channels form a large superfamily of cation channels that are essential in physiology and pathophysiology. Therefore these channels have been intensively studied. In the past few years, with breakthrough of single-particle cryoEM, near-atomic resolution structures have been obtained for members from the TRPV, TRPM, TRPP, TRPA, and TRPML subfamilies. This manuscript by Duan et al. describes the cryo-EM structure of the mouse TRPC4 channel at an overall resolution of 3.3 angstroms, representing an important structural snapshot of the last TRP subfamily (TRPC) with unknown structures until now. This work potentially provides a framework for further understanding structure and function of TRPC channels and is of broad interest to the ion channel field. The cryo-EM data and model are of good quality. In addition, functional data were included to validate the relevance of the truncated construct used for structural analysis. The authors described the TRPC4 structure reasonably well.

However, very surprisingly, the authors did not discuss structural findings in the context of any functional data available in the literature for TRPC channels. Therefore, though the new TRPC4 structure is very interesting, this manuscript is not well prepared and is boring and superficial. The manuscript could be dramatically improved by discussing the TRPC4 structure in the context of functional properties of TRPC channels, including ion permeation, gating mechanisms and mutagenesis studies that have been available in the literature. The manuscript could be much

more exciting. In addition, the authors need to be more cautious and rigorous with their writing, as exemplified in the following specific points.

Line 26: "The structure reveals an unusually complex architecture with a long pore loop stabilized by a disulfide bond." What does "unusually complex architecture" really mean?

Fig.3 How did the authors align these structures? Using the entire structures, the pore domains, or the transmembrane domains?

Line 126-130: Very confusing statement. What clues may the differences in gate dimensions give to activation mechanisms?

Line 166-169: This statement is problematic.

Reviewer #3 (Remarks to the Author):

Duan et al. presented a manuscript on the structure of the truncated mouse TRPC4 channel. This work represents a structure of the last TRP channel family member that has not been reported yet. This is important milestone for the TRP channel field, so paper is significant to grant the publication in Nature Communications. Nevertheless, I have some comments that I would like authors to address in the manuscript.

Major comments:

- 1) Currents for WT channel are not identical to the truncated TRPC4 based on the data presented in Supp Figure 1b, please add clarification in the text why currents are different between these two constructs.
- 2) In Figure 2, also some clarifications are required on why currents are different between two constructs.

Minor comments:

- 1) In Introduction (line 50-51), authors should reference all TRP channels structures determined to date, not just the one they prefer to acknowledge.
- 2) Supp Figure 1a, authors should clear show which band is corresponding to the truncated TRPC4 on the gel. It is two bands on the gel and it is not clear which one is the truncated TRPC4.
- 3) Paper reads as a comparative, descriptive work between TRPC4 and TRPM4 structures. I would recommend to make an emphasis on the TRPC4 more in the text.

Reviewers' comments:

Reviewer #1 (Remarks to the Author):

In the present manuscript, J. Duan and colleagues report the first structure of the murine TRPC4 channel at a quite high resolution of 3.3 Å. While structures of members of all other TRP channel subfamilies have already been published, a TRPC structure is still lacking. In principle, this manuscript is of outstanding interest for the scientific community.

In the meantime, the submitted manuscript and one additional TRPC4 structure and structures of TRPC3 and TRPC6 are deposited on the bioRxiv server and publicly accessible, reflecting the astounding pace with which TRP channel cryo-EM structures are produced at present.

I have only a few minor comments which may further benefit the readability of the present paper such that a broad scientific audience can profit from it.

To Reviewer 1:

Thank you very much for the positive opinion on our work and for giving us an opportunity to revise our manuscript. We appreciate your effort in this review and your constructive suggestions.

Reply to the comments:

Specific points:

(1) Abstract: Please explain what is meant by "apo" state. Maybe the authors could write: "... in its unliganded (apo) state ...".

> Thank you for your comment, we have revised "apo" state as "unliganded (apo)" state in the line new 24.

(2) Abstract, line 29: Please start new sentence: "The unique ...".

> We have revised the sentence as “This cytosolic N-terminal domain forms extensive aromatic contacts with the TRP and the C-terminal domains (line new 27-28).

(3) Introduction, line 47: Please also cite Storch et al., Proc Natl Acad Sci U S A. 2017 Jan 3;114(1):E37-E46. doi: 10.1073/pnas.1612263114. A potential mechanism underlying lipid regulation is proposed in here.

> We have added the reference and the revised text reads: “Activation of TRPC4 is regulated by intracellular Ca²⁺, phospholipase C, and membrane lipids by unclear mechanisms. In addition, Storch et al. have proposed a potential mechanism related to phosphatidylinositol 4,5-bisphosphate and Na⁺/H⁺ exchanger regulatory factor proteins¹¹ (line new 44-47).

(4) Supplementary Fig. 1b: Why were whole-cell currents in Suppl. Fig. 1b induced with trypsin? The truncated receptor inactivates within a few seconds, while the full-length channel stays activated for more than 2 min. Any explanation for this phenomenon? Can one really rule out different biophysical properties caused by the truncations?

>We used trypsin to activate TRPC4 currents because its receptor protease-activated receptor 2 (PAR2, a Gq-coupled receptor) is endogenously expressed in 293T cells. However, since the currents induced by endogenous PAR2 were quite small for both full-length and truncated TRPC4, we have instead overexpressed other GPCRs to compare the functionality of the two constructs. We found the truncated TRPC4 could only be weakly activated by the Gq-coupled receptors P2Y1 and P2Y2, and did not respond to stimulation of Gi/o-coupled receptors μ OR and M2R. These results indicate the activity of TRPC4 is dramatically affected by the C-terminal truncation. This is a drawback of our study, which has been discussed thoroughly in the revised manuscript.

(5) Figure 2: Please describe the experiments shown in Fig. 2 in the text in more detail. The reader is completely left alone to interpret the experiments shown.

> We have expanded the description in Results (line new 66-77) and in the Figure legend. Thank you for your advice.

My recommendation would be: Minor revision

Reviewer #2 (Remarks to the Author):

TRP channels form a large superfamily of cation channels that are essential in physiology and pathophysiology. Therefore these channels have been intensively studied. In the past few years, with breakthrough of single-particle cryoEM, near-atomic resolution structures have been obtained for members from the TRPV, TRPM, TRPP, TRPA, and TRPML subfamilies. This manuscript by Duan et al. describes the cryo-EM structure of the mouse TRPC4 channel at an overall resolution of 3.3 angstroms, representing an important structural snapshot of the last TRP subfamily (TRPC) with unknown structures until now. This work potentially provides a framework for further understanding structure and function of TRPC channels and is of broad interest to the ion channel field. The cryo-EM data and model are of good quality. In addition, functional data were included to validate the relevance of the truncated construct used for structural analysis. The authors described the TRPC4 structure reasonably well.

However, very surprisingly, the authors did not discuss structural findings in the context of any functional data available in the literature for TRPC channels. Therefore, though the new TRPC4 structure is very interesting, this manuscript is not well prepared and is boring and superficial. The manuscript could be dramatically improved by discussing the TRPC4 structure in the context of functional properties of TRPC channels, including ion permeation, gating mechanisms and mutagenesis studies that have been available in the literature. The manuscript could be much more exciting. In addition, the authors need to be more cautious and rigorous with their writing, as exemplified in the following specific points.

To Reviewer 2:

Thank you very much for your helpful comments. We appreciate your effort in this review and your constructive suggestions. We discuss the functional properties of TRPC channel, including ion permeation and gating, based on our structural findings in the Results and Discussion. In addition, we performed patch clamp experiments of

TRPC4 with mutations of the pore region disulfide bond.

Line 26: “The structure reveals an unusually complex architecture with a long pore loop stabilized by a disulfide bond.” What does “unusually complex architecture” really mean?

> We have revised “unusually complex architecture” as “unique architecture”. (in line new 25)

Fig.3 How did the authors align these structures? Using the entire structures, the pore domains, or the transmembrane domains?

>We aligned the entire structures. We have tried alignments of both entire structures and transmembrane domains, but there is little difference between transmembrane domain alignments among available TRPs structures.

Line 126-130: Very confusing statement. What clues may the differences in gate dimensions give to activation mechanisms?

> We have revised this sentence as “These $\sim 0.5\text{-}2.5$ Å differences in the narrowest point of TRPs structures may give some clue as to ion selectivity...” (line new 148-149).

Activation of TRPV1 is associated with structural rearrangements in the selectivity filter and the lower gate, suggesting a dual gating mechanism (Cao et al., 2013). However, for other TRP channels such as TRPA1, TRPM4 and TRPC4, the selectivity filter is large enough to allow the permeation of partially hydrated ions; opening of these channels may be regulated by the narrowest point in the lower gate.

Line 166-169: This statement is problematic.

> We have removed the sentences in (old) lines 166-169. We clarified and added more about the activation mechanisms in the Results and Discussion.

Reviewer #3 (Remarks to the Author):

Duan et al. presented a manuscript on the structure of the truncated mouse TRPC4 channel. This work represents a structure of the last TRP channel family member that has not been reported yet. This is important milestone for the TRP channel field,

so paper is significant to grant the publication in Nature Communications.

Nevertheless, I have some comments that I would like authors to address in the manuscript.

Thank you very much for the positive opinion on our work and for giving us an opportunity to revise our manuscript. We appreciate your effort for reviewing and the constructive suggestions.

Major comments:

1) Currents for WT channel are not identical to the truncated TRPC4 based on the data presented in Sup Figure 1b, please add clarification in the text why currents are different between these two constructs.

> We have overexpressed four different GPCRs with the full-length and truncated TRPC4 constructs to test the impact of C-terminal truncation. We found the truncated TRPC4 could only be weakly activated by Gq-coupled receptors P2Y1 and P2Y2, and did not respond to stimulation of Gi/o-coupled receptors μ OR and M2R. These results indicate the activity of TRPC4 is dramatically affected by the C-terminal truncation. This is a drawback of our study, which we discuss in the Discussion of revised manuscript.

2) In Figure 2, also some clarifications are required on why currents are different between two constructs.

> The amplitude and activation process of whole-cell current vary from cell to cell, especially when 1nM englerin A was used. This relatively low concentration on both the truncated and full-length constructs evokes a slow change in current and is influenced by background and leak currents. We have picked another recording as an example for the full-length TRPC4-- this shows similar activation kinetics as the truncated construct. In contrast to 1nM englerin A, the activation by 100nM englerin A is much more potent and immediate, so the recordings are more consistent between the two constructs. As shown in the dose-response curves, there is no statistical difference for the activation of full-length and truncated constructs by englerin A at different concentrations.

Minor comments:

1) In Introduction (line 50-51), authors should reference all TRP channels structures determined to date, not just the one they prefer to acknowledge.

> We have added all TRP channels structures determined to date (in line new 49-50) as read “..the structures of TRPVs¹²⁻¹⁷, TRPA1¹⁸, TRPP1¹⁹, TRPML1²⁰, TRPM4²¹⁻²⁴ and TRPM8²⁵ have been solved.”

2) Supp Figure 1a, authors should clear show which band is corresponding to the truncated TRPC4 on the gel. It is two bands on the gel and it is not clear which one is the truncated TRPC4.

> The two bands on the gel represents TRPC4 with MBP and without MBP respectively. We have clearly labeled these bands “MBP-TRPC4” and “TRPC4” in Supplementary Fig. 1.

3) Paper reads as a comparative, descriptive work between TRPC4 and TRPM4 structures. I would recommend to make an emphasis on the TRPC4 more in the text.

> We have added more structural and functional details in the Results and Discussion to emphasize TRPC4. Thank you for your advice.

REVIEWERS' COMMENTS:

Reviewer #2 (Remarks to the Author):

The manuscript is improved and acceptable for publication.

Reviewer #3 (Remarks to the Author):

My comments have been addressed. Please accept the manuscript.